# The Role of Tumor Microenvironment in Regulating the Plasticity of Osteosarcoma Cells

**DOI:** 10.3390/ijms232416155

**Published:** 2022-12-18

**Authors:** Boren Tian, Xiaoyun Du, Shiyu Zheng, Yan Zhang

**Affiliations:** MOE Key Laboratory of Gene Function and Regulation, School of Life Sciences, Sun Yat-sen University, Guangzhou 510006, China

**Keywords:** osteosarcoma, cancer stem cell, cell plastic, tumor microenvironment

## Abstract

Osteosarcoma (OS) is a malignancy that is becoming increasingly common in adolescents. OS stem cells (OSCs) form a dynamic subset of OS cells that are responsible for malignant progression and chemoradiotherapy resistance. The unique properties of OSCs, including self-renewal, multilineage differentiation and metastatic potential, 149 depend closely on their tumor microenvironment. In recent years, the likelihood of its dynamic plasticity has been extensively studied. Importantly, the tumor microenvironment appears to act as the main regulatory component of OS cell plasticity. For these reasons aforementioned, novel strategies for OS treatment focusing on modulating OS cell plasticity and the possibility of modulating the composition of the tumor microenvironment are currently being explored. In this paper, we review recent studies describing the phenomenon of OSCs and factors known to influence phenotypic plasticity. The microenvironment, which can regulate OSC plasticity, has great potential for clinical exploitation and provides different perspectives for drug and treatment design for OS.

## 1. Introduction

Osteosarcoma (OS) is a malignancy that most commonly occurs in children and adolescents and is the second highest cause of cancer-related mortality in these groups [1,2,3]. There has been a rise in the annual incidence rate of OS to three cases per million individuals [4]. The majority of OS cases arises in the metaphyseal regions adjacent to the physis, including the distal femur, proximal tibia and the proximal humerus, with a strong capacity for proliferation [5]. Over the past 30 years, the treatment of OS has improved little, such that surgery accompanied with chemoradiotherapy remain as the main method of treatment [6]. Although novel clinical strategies such as gene editing, individualized treatment and novel molecular-targeted therapies, e.g., angiogenesis inhibitors, tyrosine kinase inhibitors and monoclonal antibodies, have all been deployed against OS, the outcomes for patients are poor, particularly those with more aggressive forms of the cancers [3,4,5,6,7]. Therefore, novel treatment strategies are in demand in clinical practice. In addition, the molecular mechanism underlying tumorigenesis and malignant metastasis needs to be studied in detail.

Based on the present research, OS is speculated to have two main origins, bone mesenchymal stem cells (BMSCs) and osteoblasts [8,9,10]. p53 as a classic cancer suppressor gene plays a key role in OS progression. The deficiency of p53 is an important reason leading to primary OS. In addition, retinoblastoma gene (Rb), cyclin dependent kinase inhibitor 2 (CDKN2), KRAS and c-Met also participate in the regulation of OS progression [8,9,10,11]. Within cancer tissues, there exist several dynamic subsets of cancer cells considered to be cancer stem cells (CSCs) or stem cell-like cancer cells [12,13]. CSCs have been frequently reported to exhibit stem cell properties and capabilities of long-term clonal proliferation, tumorigenicity, facilitating metastasis and promoting resistance to chemotherapy and radiotherapy [14,15]. Therefore, exploring the origins of cancer initiation and metastasis will likely facilitate the development of future therapies. In 1994, Lapidot et.al first reported that, in human acute myeloid leukemia, a rare population of CSCs exists [16]. Subsequently, an accumulating number of studies have also reported the existence of CSCs in other solid tumors, including prostate, glioblastoma, hepatoma, breast cancer and OS [17,18,19,20,21,22]. In fact, all types of malignant tumors consist of different subpopulations of tumor cells, leading to high degrees of heterogeneity.

The niche in which CSCs reside is the tumor microenvironment, where they co-exist with adjacent supporting cells, micro-vessels and the extracellular matrix [19,20]. In addition, the tumor microenvironment can contain soluble factors, such as chemokines and cytokines, whilst being under the influence of various mechanical factors, including matrix stiffness, solid stress and fluid stress [23,24]. In the OS microenvironment, OS stem cells (OSCs) are contained in a specialized niche that contains a unique bone microenvironment, which consists of various types of bone cells, such as osteoblasts or osteoclasts. OSCs are similar to other CSCs, in that they account for a proportion of cancer cells with tumorigenic and self-renewal capabilities. The existence of OSCs was first confirmed by Gibbs et al., who found that when primary human OS cells or the OS cell line MG63 were suspended in a serum-free medium with defined growth factors, 0.1% of the cells could form spheres with self-renewal capacity [17,18,19,20,21,22,23,24,25]. Subsequently, a series of studies have proven the existence of OSCs, in addition to revealing the phenotype and possible marker profile of OSCs. Here, we summarize and review the recent studies on possible OSC markers and phenotypes (Table 1).

The primal CSC theory is a hierarchical model that proposes that CSCs have similar patterns compared with normal stem cells. According to this theory, CSCs can either undergo asymmetric fission, where they can divide to give two different progenies (one CSC and the other non-CSC), or symmetrically into two CSCs [43]. CSCs have shown great potential as a target for cancer chemotherapy according to this hierarchical model. Stem cell research strategies have been previously applied for the exploration of CSCs, specifically of the specific markers of CSCs [44,45]. However, thus far it has not been possible to identify a marker that can be used to definitively identify cancer cells with ‘stem’ characteristics in distinct tumors and tumor cell lines. To date, there is no special marker that can accurately represent or identify CSCs in any cancer type or cell line. Previous studies have even suggested that mature leukemic progenitor cells or cells expressing lineage markers can initialize tumorigenesis [46,47]. This hierarchic model has been questioned. Accordingly, a new stochastic theory of tumorigenesis has been proposed. The new theory states that each cancer cells can switch phenotypes to gain the CSC phenotype under certain conditions in the microenvironment. According to this theory, non-CSCs can transform into CSCs if certain parameters in the tumor microenvironment are met [48,49,50]. Indeed, a series of studies have previously demonstrated that, under certain conditions or after eradication of CSCs, some non-CSCs can transform to gain the CSC phenotype to facilitate tumor progression [51,52,53].

Importantly, since tumors are comprised of various different cell types and not only cancer cells, but this inter-conversion does also not only affect CSCs and non-OSCs. In fact, the tumor microenvironment is highly complex and can contain other cell types, such as immune cells and cancer-associated fibroblasts. It can also contain soluble factors and mechanical components, such as growth factors. In addition, the tumor microenvironment can come under the influence of various stimuli, such as hypoxia. In this review, we summarize the research describing the phenomenon of cancer cell reversion, especially for OS. Furthermore, the factors that have been shown to influence cell plasticity switch are mentioned. The mechanism underlying this reversion, the influence of the tumor microenvironment composition and their overall effect on the metastatic spread of the disease are also discussed.

## 2. Role of the Tumor Microenvironment in Regulating OSC Stemness

OSCs can interact with their microenvironment through complex and dynamic processes, including variation of oxygen, mechanical interactions, enzymatic modification of the extracellular matrix (ECM) structure and signaling cross-talk, all of which can influence the progression and the dissemination of OS cells (Figure 1)

### 2.1. Hypoxia

Common hallmarks of solid tumors include intra-tumoral hypoxia, necrosis, acidic environments and disturbed angiogenesis. Previous studies have shown that increased cancer cell stemness is associated with intra-tumoral hypoxia [54,55,56,57]. Our previous study demonstrated that a hypoxia microenvironment could induce non-OSCs dedifferentiation into OSCs by increasing the expression of TGF-β [58].

During the process of tumor formation, excessive proliferation of cancer cells consumes large quantities of oxygen in the microenvironment, resulting in the formation of a hypoxic zone in the central area of the tumor. In addition, aberrant secretion of angiogenetic factors, including vascular endothelial growth factor A and fibroblast growth factor 2 (FGF2), result in malformation and disorder in the neovascularization system [54,55]. This in turn causes the loss of oxygen supply, further aggravating hypoxia in the cancer tissue [59,60].

This hypoxic microenvironment induces the expression of hypoxia-inducible factor-1 (HIF-1), a vital member of the HIF family. By contrast, in the presence of oxygen, HIF-1 undergoes degradation by the von Hippel-Lindau protein, a tumor suppressor protein [61,62,63,64,65]. HIF-1 is a heterodimer that is ubiquitously expressed in human and mouse tissues. HIF-1 consists of two subunits, the hypoxia-inducible, oxygen-dependent subunit HIF-1α and the constitutively-expressed oxygen-independent subunit HIF-1β [66,67,68]. It is only when the oxygen concentration reaches <5% (such as when the volume of the tumor has grown to >300 mm^2^) that HIF-1α can exist stably. Activity of HIF-1 provides cancer cells with the ability to adapt to hypoxia and is closely associated with tumor metabolism, differentiation, angiogenesis, cell proliferation, metastasis and multidrug resistance. Of note, several studies have demonstrated that elevated expression of HIF-1α promoted the dedifferentiation of cancer cells into CSCs, whereas hypoxia is directly associated with poorer prognoses in patients with OS [69].

Numerous studies have demonstrated that hypoxia can promote the expression of the stem cell marker CD133 to maintain stemness and drug resistance in the Saos-2 OS cell line [70]. Lin et al. previously reported that hypoxia can increase the expression of embryonic stem cell markers, including Oct3/4 and Nanog, in the MNNG/HOS OS cell line [71]. Zhang et al. showed that a hypoxic microenvironment stabilized HIF-1α in OS cells, such that HIF-1 promoted the expression of microRNA (miR or miRNA)-210, which then induced and accelerated the dedifferentiation of OS cells into OSCs [58,59,60,61,62,63,64,65,66,67,68,69,70,71,72]. These observations aforementioned suggest that HIF-1 and subsequent hypoxia signaling pathways can regulate the differentiation of CSCs and the dedifferentiation of non-stem cells in tumors [73,74].

In addition to HIF, hypoxia can also cause integrin-linked kinase dysfunction, triggering CSCs formation [75]. Hypoxia has been previously found to promote breast cancer stemness by HIF-dependent and AlkB homolog 5-mediated N6-methyladenosine (m6A)-demethylation of Nanog mRNA [76]. Shi et al. used the evolutionary theory to identify the hypoxic adaptation-associated gene YTH N6-methyladenosine RNA binding protein 1 (YTHDF1). As a member of the N6-methyladenosine (m6A)-modified RNA-binding protein family, YTHDF1 may interplay with other m6A modifiers and serve a pivotal role in the self-renewal and differentiation of stem cells [77]. Under hypoxia, AKT will accumulate in the mitochondria of tumor cells, whereby 3-phosphoinositide-dependent protein kinase 1 is phosphorylated at special sites. This pathway shifts the tumor metabolic program to glycolysis, which antagonizes apoptosis and autophagy and inhibits oxidative stress. This in turn maintains the survival and proliferation capabilities of tumor cells, as evidenced by the sphere-forming ability of cells in 3D cultures under severe hypoxia [78].

These aforementioned findings suggest that hypoxia may contribute to the creation of a microenvironment rich in tumor stem cells, where this unique hypoxic microenvironment may provide essential cellular interactions and environmental signals for the maintenance of CSCs [70,71,72,73,74,75,76,77,78,79]. By contrast, the hypoxia microenvironment can also regulate non-CSC dedifferentiation by regulating the activities of other pathways, including epithelial-mesenchymal transition (EMT), metabolic reprogramming, DNA hypermethylation and apoptotic resistance. Additionally, the hypoxic microenvironment can mediate the resistance of CSCs against drugs through drug transporters [80]. The majority of CSCs express the ATP-binding cassette (ABC) family of membrane transporters at high levels, including multidrug resistance gene 1, breast cancer resistance protein and multidrug resistance-associated protein. These proteins can transport metabolites, drugs and other substances, allowing CSCs to become highly resistant to chemotherapy. The relationship between hypoxia and ABC proteins was previously documented to have a strong association with mediating tumor drug resistance [81].

In conclusion, the hypoxic microenvironment with the activation of hypoxic signaling can serve key roles in the dedifferentiation of OS cells into OSCs. Therefore, it is important to study the molecular mechanism underlying OS dedifferentiation, which is expected to hold important clinical significance for improving the efficacy of therapeutic strategies. However, the molecular mechanism of how exactly the hypoxic microenvironment can regulate OSC physiology biology requires additional experimental evidence for validation. In particular, HIF-1 is a key molecule of the hypoxia signaling pathway, the downstream molecules of which are expected to become important markers and potential molecular targets of OSCs.

### 2.2. Biomechanical Force

Under physiological conditions, most if not all organisms experience complex biomechanical forces, including shear stress, matrix stiffness, tension and compression pressure [82,83,84,85,86]. Biomechanical forces experienced by solid tumors have different profiles compare with those in the surrounding or healthy tissue [87]. Throughout the process of cancer development, excessive cell proliferation will lead to the abnormal development of the biomechanical microenvironment, including solid stress, increased matrix stiffness (decrease in OS due to osteolysis) and abnormal interstitial fluid pressure [88,89,90].

These complex mechanical systems are essential for the maintenance of the homeostasis of the CSC population. Indeed, previous studies have demonstrated that non-CSCs can be transformed into CSCs by receiving mechanical signals from the surrounding microenvironment, such as increased matrix stiffness [91,92,93,94] and/or fluid shear stress [95,96,97,98]. In OS, soft substrate (7 kPa) has been reported to preserve OS stemness, mainly through miR-29b/Spin 1-dependent signaling. Manipulation of cancer niche stiffness and miR-29b expression may therefore be potentially novel drug targets in OS [99]. Previous studies have shown that EMT can promote the progression and invasion of tumors [100]. EMT have been observed to serve as a direct link between non-CSCs and the gain of CSC properties [101]. Matrix stiffness in the tumor microenvironment can actively regulate EMT and migration of OS cells through cytoskeletal remodeling and the translocation of myocardin related transcription factor A, which may contribute to cancer progression [102]. Although the aforementioned studies revealed that mechanical factors are at least partially associated with the dynamic conversion between non-CSCs and CSCs, further research into the association between mechanical signaling and OSC stemness is warranted.

Mechanical receptors on the cell surface, such as integrins, CD44 and ion channels, can sense the changes in ECM and activate key downstream molecules, including focal adhesion kinase, integrin-linked kinase, RhoA and yes-associated protein. Several of the signals induce non-CSC reprogramming and transform them into CSCs by increasing the expression of sex determining region Y-box 2 (Sox2), octamer-binding transcription factor (Oct)-4 and Nanog [91,92,93,94,95,96,97,98,99,100,101,102,103,104]. CSCs and normal stem cells frequently share similar surface markers and signaling pathways, which would restrict the design of treatment regimens [105]. The abnormal mechanical system in OS microenvironments, which rarely occur in the harmonious microenvironments of normal stem cells, may provide novel insights for designing CSC-targeted treatment methods. As such, discovering the relationship between biomechanical factors and CSCs will greatly enable the generation of novel research strategies to investigate the occurrence, development, and recurrence of cancers.

### 2.3. Growth Factors

Growth factors are pivotal in maintaining the physiological behavior of healthy individuals. Cells in the tumor microenvironment can secrete growth factors to regulate processes of tumor development [106,107]. When OS arise in the bone, OS cells secrete factors that direct osteoclast-mediated bone destruction. In addition, matrix-derived growth factors, especially transforming growth factor β1 (TGF-β1), are released from bone matrix. In addition, OS cells can release TGF-β1 directly, where increased TGF-β1 expression is associated with high-grade metastases of OS [108]. TGF-β1 is a multi-function cytokine that serves as a mediator in the tumor to facilitate further tumor expansion, metastasis and cytokine production [109]. Wang et.al previously reported that TGF-β1 can switch the OSC chemoresistance through the miR-499a/SHKBP1 axis [110]. In another study, TGF-β1 signaling and a hypoxic environment were found to induce the transformation of non-OSCs into OSCs dynamically, which promoted the acquisition of chemoresistance, tumorigenicity, neovasculo-genicity and metastatic potential. Furthermore, blocking the TGF-β1 signaling pathway was reported to inhibit this switch from non-OSCs to OSCs, inhibit OSC self-renewal and suppress hypoxia-mediated dedifferentiation [58]. In the bone microenvironment, TGF-β1 signaling is responsible for OSC generation and critical to chemoresistance in vivo. In addition to OS, TGF-β1 can also regulate the dynamic switching between stem cells and non-stem cells to influence the progression of tumors from different tissue origins [111,112]. In conclusion, TGF-β1 serves a key role in regulating the dynamic plasticity of OSC, which can lead to non-stem cells adopting OSC characteristics to promote tumorigenesis and chemoresistance, highlighting TGF-β1 as a potential therapeutic target.

Bone morphogenetic proteins (BMPs) are members of the TGF-β superfamily and serve important roles in the activity of various tissues. In OS, BMP-2 suppresses tumor growth by reducing the expression of oncogenes whilst promoting the differentiation of OSCs [113]. Histological examination and gene expression analysis of OS tissues revealed that fibrotic remodeling of the tumor microenvironment favors tumorigenesis. Zhang et al. previously demonstrated that fibrotic reprogramming in the lung induced by OSCs is critical for OS pulmonary metastasis, with FGF-FGF receptor 2 (FGFR2) signaling being responsible for this important process [114]. In OS, the tumor necrosis factor-α/miR-155 axis has been found to induce OSC transformation between non-OSCs and OSCs through the extracellular signal-regulated protein kinase signaling pathway [115]. Melatonin, one of the hormones secreted by the pineal gland of the brain, has been shown to significantly inhibit sphere formation by OSCs through the key transcription factor Sox-9 [116]. Although all of the aforementioned growth factors have shown the potential to target OSCs, the underlying mechanism require further exploration.

### 2.4. Cancer-Associated Cells

Together with intrinsic tumor cell changes, other cell types in the surrounding microenvironment, including fibroblasts, endothelial cells, immune cells and mesenchymal stromal cells (MSCs), can all contribute functional variety by interacting with the tumor cells [117]. MSCs within the microenvironment have been known to secrete a number of cytokines. It has been frequently reported that cytokine production by tumor-associated stroma can stimulate tumor sustenance, growth and angiogenesis, where in OS this is no exception. In addition, several reports have previously found that MSCs can produce soluble factors to regulate cancer cell stemness [118]. In particular, MSCs can secrete TGF-β1 and interleukin (IL)-6, which in turn increases stemness, cell proliferation, migration and the metastatic potential of OSCs [115]. Towards OSCs, MSCs can also increase the expression of adhesion molecules, such as intercellular adhesion molecule-1 [119].

The immune component of the OS microenvironment is mainly comprised of tumor-associated macrophages (TAMs), myeloid-derived suppressor cells, dendritic cells and regulatory T cells. Macrophages can engulf and digest foreign substances to clear potentially harmful material, such as tumor cells [120]. M2 macrophages were found to be enriched in OS tissues. M2-type TAMs have been reported to promote OS cell stemness by upregulating the expression of stemness markers whilst facilitating colony formation, sphere formation and tumor initiation [121]. All-trans retinoic acid treatment has been documented to prevent the M2 polarization of TAMs, which then diminished the CSC phenotypes, including colony- and sphere-forming capabilities [121]. TAMs have also been shown to reinforce the CSC populations through direct interactions between ephrin and ephrin type A receptor 4. This leads to the production of inflammatory cytokines IL-1, IL-6 and IL-8 by CSCs, which sustains the CSC state [122,123]. However, the mechanism by which TAMs can upregulate the CSC-like phenotype in OS remains unknown.

It is becoming accepted that fibrotic remodeling of the tumor microenvironment generally favors tumorigenesis. Myofibroblasts synthesize and deposit matrix fibrils into the extracellular space, which is one of the hallmarks of fibrosis. A previous study demonstrated that inducing fibrotic reprogramming in OSCs is critical for the growth of lung metastases. Fibronectin auto-deposition has been observed to sustain fibro-genic reprogramming and OSC proliferation, which resembles the process that occurs when non-malignant myofibroblasts induce tissue fibrosis [114].

Complex cell-to-cell interactions are necessary for maintaining the homeostasis of the CSC population. All cellular components in the tumor niche, including CSCs, non-CSCs, fibroblasts, immune cells and mesenchymal cells, can perceive the modulator effects and soluble proteins emitted by CSCs, resulting in non-CSC and CSCs interconversion. In turn, other cell types in the tumor microenvironment activate paracrine signaling pathways to ensure CSC proliferation and dissemination. However, there remains numerous as yet unknown mechanisms mediated by tumor-associated cells in the microenvironment that can regulate the plastic transition between non-CSCs and OSCs.

## 3. Extracellular Vesicles (EVs)

EVs are small membranous vesicles released by cells into the extracellular matrix. EVs are abundant throughout in the body and can stably carry and transfer important signaling molecules between cells, serving as another mechanism of cell–cell communication [124,125]. Accumulating evidence has shown that EVs may also be important in regulating OS development, progression and metastasis, supporting the notion that EVs can be of use as potential biomarkers for the diagnosis and prognosis of OS [126,127,128,129].

Previous studies have reported the potential biological role of exosomes in CSCs. Yang et.al found that human umbilical vein endothelial cell (HUVEC)-derived exosomes enhanced the proportion of CD117+ cells relative stemness gene, which increased sphere formation. Furthermore, HUVEC-exosomes have been found to promote cell stemness in OS by activating the Notch signaling pathway [130]. Zhang et.al demonstrated that bone mesenchymal stem cell (BMSC)-derived EVs could be transferred into OS cells to inhibit tumor progression by targeting transformer 2β homolog (TRA2B), before subsequently proposing the potential of miR-206 and TRA2B as novel therapeutic targets [131]. In another previous study, BMSC-derived EVs were found to activate the miR-30-5p/Kruppel-like factor 10 axis in OS cells to promoting cell proliferation and lung metastasis [132].

In pancreatic cancer, EVs have been observed to exert significant effects on regulating CSCs. M2 macrophage-derived EVs have been documented to promote pancreatic cancer stem cell differentiation and activities through miR-21-5p [133]. EVs derived from chemo-sensitive non-stem bladder cancer cells were found to be enriched with cargo proteins that can mediate proteo-static functions to significantly alter the development of CSCs [130]. As a result, they became more intrinsically chemo-resistant and aggressive with enhanced self-renewal capabilities [134]. Furthermore, EVs derived from human liver stem cells or MSCs have been found to reduce CSC proliferation and invasion whilst increasing CSC apoptosis, but had no effect on non-CSCs [135].

Due to their biogenesis, EVs may contain a high variety of molecular cargoes that is dependent on their cell of origin. DNA, proteins, mRNA, microRNA and lipids, all of which can regulate signaling pathways inside target cells, have been documented [136]. There is ample evidence that CSC-derived EVs can promote stem-like properties in non-CSCs, leading to the enhanced tumorigenicity [137,138,139]. For these reasons, EVs are now considered to be leading facilitators in promoting the dynamic interconversion between non-CSCs and CSCs. However, in OS, experimental research data on the composition of EVs in OSCs remaining lacking at present. In addition, the components within EVs secreted by different cell types in the OS microenvironment remain obscure. Further research is required to characterize the cellular and noncellular components of the OS microenvironment to understand how it regulates OSC plasticity.

## 4. Non-Coding RNAs (ncRNAs) in Regulating OSCs

Developmental signaling pathways, including the Notch, Wnt, Hedgehog and Hippo pathways, are commonly found to be dysregulated in CSCs. Indeed, these signaling pathways can all serve key regulatory functions that support the maintenance and survival of CSCs. Further research is needed to deepen the understanding into these signaling pathways aforementioned in OSCs. In recent years, research into the roles of non-coding RNAs in OSCs has been on the increase. A series of studies have previously shown that (ncRNAs) can regulate the dynamic OSC plasticity (Figure 2) [140]. ncRNAs are RNA sequences that do not encode proteins but can regulate gene expression, which include micro miRNAs (miRNAs), long ncRNAs (lncRNAs) and circular RNAs (circRNAs) [141]. miRNAs are typically 18–25 nucleotides in length and target specific mRNAs by either complete or partial complementary binding to their 3′untranslated regions (UTR) [142]. By contrast, lncRNAs are >200 nucleotides in length and mainly exert their functions by sponging miRNAs and targeting specific substrates [143].

### 4.1. miRNAs

In recent years, miRNAs have been reported to regulate CSCs in a variety of cancers, including OS [144]. Research in our laboratory has shown that miR-34a serve a key role in regulating non-OSC dedifferentiation into OSCs. The miR-34 could inhibit OS dedifferentiation into OSCs through the Sox2-PAI pathway [145]. miR-26a expression has been found to be reduced in OSCs, such that miR-26a overexpression was able to inhibit sphere formation and tumor cell proliferation both in vitro and in vivo. miR-26a can also inhibit OS cell stemness by targeting Jagged1 expression, one of the Notch ligands [146]. In another study, Zou et.al previously revealed that miR-34a expression was lower in OSCs, where overexpression of miR-34a reduced sphere formation ability and the expression stem cell marker genes [147]. Liang et.al also suggested that increased miR-34a expression in OS can inhibit sphere and colony formation abilities [148]. In addition, Zhao et.al reported that miR-1247 can inhibit CD117+ CSC sphere formation and stem cell-associated gene expression [149]. Patients with OS and lower expression levels of miR-382 were associated with poorer chemotherapy responses and poorer prognoses, whilst upregulating the expression of miR-382 was found to inhibit pulmonary metastasis and reduced the population of OSCs [150]. Increasing miR-29b-1 expression was also found to increase drug susceptibility in OS cells. In addition, miR-29b-1 overexpression could also inhibit the expression of stem cell genes by suppressing Oct3/4, Sox2 and Nanog in OSCs [151]. In another previous study, Guo et.al reported that OSCs have lower miR-335 expression levels, because miR-335 inhibited OS cell stemness by suppressing the Oct-4 pathway [152]. Overexpression of the miRNA lethal-7 has been reported to reduce sphere formation and decrease the expression of stem cell markers in OSCs.

### 4.2. lncRNAs

Recently, a number of studies have reported that lncRNA can alter the balance between non-OSCs and OSCs. One previous study found that expression of the lncRNA differentiation antagonizing non-protein coding RNA (DANCR) was increased in OS tissues and OS cell lines. DANCR can also enhance tumor malignancy by increasing the OSC pool through activation of the PI3K/AKT signaling pathway in OS [153]. LncRNA DLX6-antisense 1 (AS1) overexpression in OS was reported to promote stemness in OS cells through miR-129-5p activating Wnt pathway. Zhang et.al. also found that higher expression levels of DLX6-AS1 in OS tissues tended to associate with poorer prognosis [154]. In addition, overexpression of LncRNA hypoxia-inducible factor-2α promoter upstream transcript decreased the population of CD133+ OSCs, which inhibited the sphere-forming capacity [155]. LncRNA testis-associated oncogenic lncRNA was shown to directly bind to the 3′-UTR of SOX9 mRNA to increase stemness in the OS cell line MG-63 [156]. Ma et.al. revealed that inhibition of lncRNA fer-1 family member 4 upregulated the expression of stemness related marker [157]. LncRNA long intergenic non-coding RNA for kinase activation was found to at least in part govern the stemness of OS by decreasing the percentage of CD133+ cells in OS cell lines [158]. Chen et.al. reported that lncRNA metastasis-associated lung adenocarcinoma transcript 1 (MALAT1) is highly expressed in tumor tissues and is associated with tumor size, metastasis and poorer survival in patients with OS. Furthermore, MALAT1 has been discovered to regulate stem cell expression in OS [159]. LncRNA SOX2 overlapping transcript (OT) was found to be overexpressed in OS cell lines and patients with OS, where knocking down the expression of this lncRNA could significantly decrease the expression of stemness biomarker [160]. Li et.al. demonstrated that lncRNA β-1,4-galactosyltransferase 1-AS1 knockdown inhibited sphere formation and decreased stemness marker expression [161]. Shi et.al. reported that circRNA phosphatidylinositol-4-phosphate 5-kinase type 1α could significantly inhibit sphere formation by OS cells and decreased the CD133 + /CD44+ cell population [162].

## 5. Targeting the Microenvironment of OSCs

The complex interactions among various components and tumor cells within a specific microenvironment serves a critical role in tumor growth and dissemination. Cancer cells require mechanical support, blood supply and growth factors secreted by cancer associated cells to guarantee continuous growth [163]. The crosstalk between OSCs and the microenvironment provides a variety of potential therapeutic strategies for the targeted treatment of OSCs, in theory. However, OSCs can resist apoptosis using various methods [164], allowing them to survive the clinical treatments currently available [10]. In addition, a number of treatment strategies may even induce non-OSC transformation into CSCs [165].

TGF-β1 as a key component of the tumor microenvironment and can regulate the interconversion between CSC and non-CSCs, suggesting that it has potential as a treatment target. A strategy combining a nanoparticle-based vaccine with the targeted silencing of TGF-β expression using liposome-protamine-hyaluronic acid nanoparticles has been previously attempted. Since TGF-β can function as an immune suppressor, silencing it may enhance the systemic immune response against the tumor [166]. Fresolimumab (GC-1008) as TGF-β inhibitor is used to treat CSCs and GC-1008 IN PhaseⅠorⅡ, correspondingly for malignant melanoma or metastatic breast cancer [167]. Besides, inducing CSC differentiation is also an efficient strategy and a series of drugs aiming Wnt or Notch signal pathway also have great therapy potential [168,169]. Notch signaling constitutes a highly conserved cell fate determining pathway with functions pertinent to a wide breadth of cancer biology, including the CSC phenotype, angiogenesis, metastasis and tumors immune evasion. A new drug, rovalpituzumab tesirine (Rova-T) conjugated DDL3 (an atypical Notch ligand) antibody, is performing in Phase Ⅲ for small-cell lung cancer [170]. Salinomycin can selectively target OSCs to inhibit OS cell proliferation through the Wnt/β-catenin signaling pathway [171]. In addition, other attractive CSC immunotherapeutic targets supported by preclinical data include chemokine receptors, such as the IL-8 receptor C-X-C motif chemokine receptor 1, IL-8 and IL-6 [172]. Furthermore, IL-6 and IL-8 inhibitors tocilizumab and reparixin have been demonstrated to show potential in chemotherapy.

Although targeting OSCs is considered to be one of the more promising research fields as a novel treatment strategy, the majority of OSC-based treatments have not been able to successfully enter clinical trials. A variety of reasons have been proposed for this suboptimal translation from bench to bedside, such as inadequate physicochemical features of OSCs and scarce knowledge of the interconversion mechanism.

## 6. Conclusions

According to the dynamical OS cell plasticity, the most promising strategies for preventing OS metastasis should be those that can target the activator of the OSCs instead of the OS itself. This is because of the potentially dynamic swing between OSCs and non-OSCs. Deciphering the OS microenvironment would lead to a deeper understanding of the fundamental nature of the OSC/non-OSC crosstalk, the interconversion between non-OSCs and OSCs and ultimately their impact on future clinical treatment outcomes. Likewise, exploring OS cell reprogramming occurring in the TME should also open the possibility of designing novel strategies to combat OS relapse and metastatic spread.

## Figures and Tables

**Figure 1 ijms-23-16155-f001:**
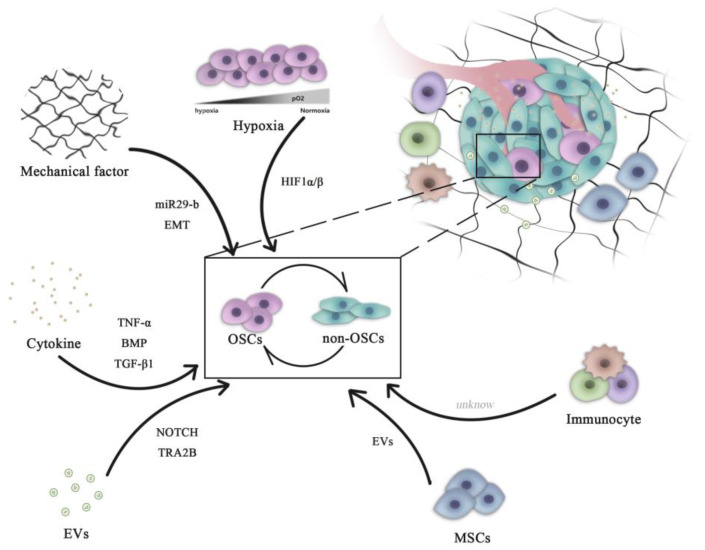
Role of microenvironment signaling in regulating OSC and non-OSC reversion. Complex pathways are necessary for the maintenance of the homeostasis of the OSC population. All microenvironment components, including cells (mesenchymal cells and immune cells) and non-cellular factors (hypoxia, cytokines and mechanical EVs), can influence the dynamic transition between OSCs and non-OSCs. OSCs, osteosarcoma stem cells; miR, microRNA; HIF, hypoxia-inducible factor; EVs, extracellular vesicles; EMT, epithelial-mesenchymal transition; BMP, bone morphogenetic protein; TNF, tumor necrosis factor; TGF, transforming growth factor; TRA2B, transformer 2β homolog.

**Figure 2 ijms-23-16155-f002:**
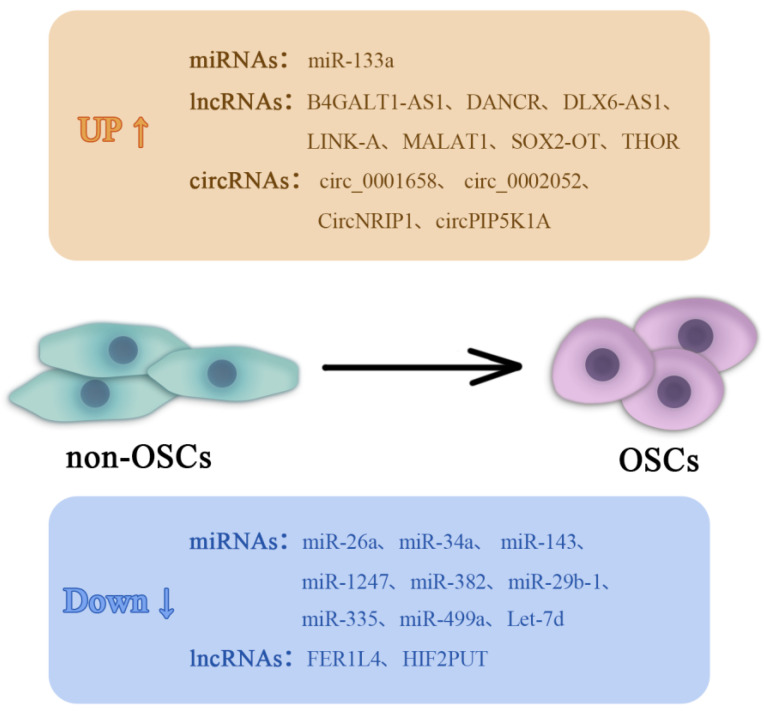
Key non-coding RNAs in non-OSC and OSC interconversion signaling. A series of studies have reported that a number of non-coding RNAs, including microRNAs, lncRNAs and circRNAs, can participate in the OSC interconversion mechanism. OSC, osteosarcoma stem cell; lncRNA, long non-coding RNA; circRNA, circular RNA; AS1, antisense 1; B4GALT1, β-1,4-galactosyltransferase 1; DANCR, differentiation antagonizing non-protein coding RNA; LINK-A, long intergenic non-coding RNA for kinase activation; MALAT1, metastasis-associated lung adenocarcinoma transcript 1; Sex-determining region Y-box 2 overlapping transcript; THOR, testis-associated oncogenic lncRNA; NIRP1, Nuclear receptor interacting protein 1; PIP5K1A, Phosphatidylinositol-4-Phosphate 5-Kinase Type 1^α^; let, lethal; FER1L4, fer-1 family member 4; HIF2PUT, hypoxia-inducible factor-2α promoter upstream transcript.

**Table 1 ijms-23-16155-t001:** Putative OSC markers and phenotypes.

Marker	Cell Origin	Phenotype
CD133	Saos-2, MG-63, U2-OS, MNNG/HOS, 143B, HOS, Human primary cells	High stem cells gene expression, sphere formation, side population, increased cell proliferation [26,27,28,29,30].
CD117/Stro-1	K7M2, KHOS/NP, MNNG/HOS, 318–1, P932, BCOS	High stem cells gene expression, sphere formation, drug resistance, in vivo tumorigenicity and metastatic potential [17,18,19,20,21,22,23,24,25,26,27,28,29,30,31].
CD271	Human primary (FFPE), MNNG/HOS, U2-OS, Saos-2	High stem cells gene expression, sphere formation, drug resistance, in vivo tumorigenicity [32].
Aldehyde dehydrogenase	MG-63, OS99–1 Hu09, Saos-2	High stem cells gene expression, sphere formation, drug resistance, increased cell proliferation [33,34].
Stem cells antigen-1	4 Murine osteosarcoma cell lines	Sphere formation, in vivo tumorigenicity [35,36]
Fas apoptotic inhibitory molecule 2	MNNG/HOS, U2-OS	Sphere formation, drug resistance, in vivo tumorigenicity [37].
Side population	OS2000, KIKU, NY, Huo9, HOS, U2OS, Saos-2, human primary	High stem cells gene expression, Sphere formation, in vivo tumorigenicity, self-renewal, apoptosis resistant [38,39,40].
Sphere formation	MG-63, MNNG/HOS, human primary	High stem cells gene expression, drug resistance, in vivo tumorigenicity [17,18,19,20,21,22,23,24,25,26,27,28,29,30,31,32,33,41,42].

## Data Availability

Not applicable.

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
