# Peer review of "The Role of Tumor Microenvironment in Regulating the Plasticity of Osteosarcoma Cells"

_ijms, 2022, doi:10.3390/ijms232416155_

Round 1
Reviewer 1 Report
The manuscript entitles "The role of tumor microenvironment in regulating the plasticity of osteosarcoma cells " summarize the phenomenon of OSCs and factors known to influence the phenotypic plasticity of osteosarcoma cells. it is a detailed review of existing literature. However, I still have several concerns:
1. It's not clear the cell origin and genetic/epigenetic aberration of osteosarcoma cells.
2. Osteosarcoma is a type of bone cancer, do the microenvironment in osteosarcoma have a different microenvironment from bone metastasis tumor
3. It would be better if you can add a graph summary for the microenvironment of OSCs and highlight the important factors in the osteosarcoma tumor microenvironment
4. Is there any drug or ongoing clinical trial targeting osteosarcoma tumor microenvironment or osteosarcoma cells plasticity
Author Response
Response to Reviewer 1 Comments
Point 1: It's not clear the cell origin and genetic/epigenetic aberration of osteosarcoma cells
Response 1: Thank you for your interest in this part. Based on the present research, osteosarcoma cells have two main origins, bone mesenchymal stem cells (BMSCs) and osteoblasts. p53 as a classic cancer suppressor gene plays a key role in the osteosarcoma progression. The deficiency of p53 is an important reason leading to primary OS. In addition, retinoblastoma gene (Rb), cyclin dependent kinase inhibitor 2 (CDKN2), KRAS, c-Met, also participate in the regulation of OS progression. According to your opinion, we added the corresponding description in manuscript page 1 and line 39-44. (in yellow)
Point 2: Osteosarcoma is a type of bone cancer, do the microenvironment in osteosarcoma have a different microenvironment from bone metastasis tumor
Response 2: Thank you for your comment. Most of primary osteosarcoma and bone metastasis cancer, such as breast cancer, prostate cancer, lung cancer, thyroid cancer and kidney cancer are involved in hypoxic and acidic microenvironment [1,2]. In addition, primary osteosarcoma and bone metastasis cancer also share many chemokines and cell types in the bone and bone marrow microenvironment, including osteoblasts, osteoclasts, osteocytes, fibroblasts and immune cells, which have been proved to affect the progression and malignancy of OS and bone metastasis cancer [3].
On the other hand, there are some differences. For example, several studies demonstrated that cancer associated fibroblasts (CAFs) originating from primary cancers could travel with metastatic cancer cells, which participate in the remodeling of metastatic microenvironment [4,5]. Also, matrix metalloproteinases (MMPs) and C-C chemokine ligand 5 (CCL5) are highly expressed in bone metastatic prostate cancer [6], while high level of adenosine nucleotides is found in bone metastatic breast cancer [7]. All of these are the tips for us, osteosarcoma and different types of bone metastasis cancers may have different microenvironment, more research is required for further elucidation.
References:
[1] C. Yang, Y. Tian, F. Zhao, Z. Chen, P. Su, Y. Li, A. Qian, Bone Microenvironment and Osteosarcoma Metastasis, Int J Mol Sci 21 (2020). 10.3390/ijms21196985.
[2] W. Zeng, R. Wan, Y. Zheng, S.R. Singh, Y. Wei, Hypoxia, stem cells and bone tumor, Cancer Lett 313 (2011) 129-136. 10.1016/j.canlet.2011.09.023.
[3] D. Buenrostro, P.L. Mulcrone, P. Owens, J.A. Sterling, The Bone Microenvironment: a Fertile Soil for Tumor Growth, Curr Osteoporos Rep 14 (2016) 151-158. 10.1007/s11914-016-0315-2.
[4] D.G. Duda, A.M. Duyverman, M. Kohno, M. Snuderl, E.J. Steller, D. Fukumura, R.K. Jain, Malignant cells facilitate lung metastasis by bringing their own soil, Proc Natl Acad Sci U S A 107 (2010) 21677-21682. 10.1073/pnas.1016234107.
[5] P.J.B. Matthew Kraman , James N. Arnold , Edward W. Roberts , Lukasz Magiera , James O. Jones , Aarthi Gopinathan , David A. Tuveson, Douglas T. Fearon Suppression of Antitumor Immunity by Stromal Cells Expressing Fibroblast Activation Protein–α., Science (2010).
[6] J.L. Sottnik, J. Dai, H. Zhang, B. Campbell, E.T. Keller, Tumor-induced pressure in the bone microenvironment causes osteocytes to promote the growth of prostate cancer bone metastases, Cancer Res 75 (2015) 2151-2158. 10.1158/0008-5472.CAN-14-2493.
[7] M.A.R. J Z Zhou, X Gao, L G Ellies, L Z Sun, J X Jiang Differential impact of adenosine nucleotides released by osteocytes on breast cancer growth and bone metastasis, Oncogene (2015).
Point 3: It would be better if you can add a graph summary for the microenvironment of OSCs and highlight the important factors in the osteosarcoma tumor microenvironment
Response 3: Thank you for comment. To highlight the important factors in osteosarcoma microenvironment, we drew the Figure.1 that could display the main component of tumor microenvironment. In the Figure.1, we summarized the interaction between microenvironment and OSCs. During the dynamic change between OSCs and non-OSCs, components in microenvironment such as EVs, cytokine could active or inhibit the corresponding signals to promote the cancer progression.
Point 4: Is there any drug or ongoing clinical trial targeting osteosarcoma tumor microenvironment or osteosarcoma cells plasticity
Response 4: Thank you for your comment. Several studies reported the drugs targeting CSCs. Fresolimumab (GC-1008) as TGF-β inhibitor is used to treat CSCs and GC-1008 is going on Phaseâ… or â…¡ corresponding for malignant melanoma or metastatic breast cancer. Another drug is rovalpituzumab tesirine (Rova-T) conjugated DDL3 (an atypical Notch ligand) antibody performing in Phase â…¢ for small-cell lung cancer. Besides, inducing CSCs differentiation is also an efficient strategy, a series of drugs aiming Wnt or Notch signal pathway also have great therapy potential. According to your helpful advice, we added corresponding details in the manuscript. (Page 11 line 436-440 and line 443-445, in yellow)

Reviewer 2 Report
Review_ijms-2056467, Tian et al. reviewed recent basic research study on the tumor microenvironment regulating cancer stem cell plasticity in the setting of osteosarcoma. The author summarized the factors that potentially contribute to OSC plasticity leading to cancer therapeutic resistance, relapse and metastasis. However, this review is more tend to be literature summarization without attractive opinions and perspectives on the shaping of cancer stem cell plasticity. Beyond that, the author should combined with their own studies on cancer stem cell plasticity to demonstrate review opinions. More importantly, if it is available, how about clinical practice on targeting cancer stem cell, and can we apply experiences from other cancer types on the cancer stem cell plasticity?

Author Response
Response to Reviewer 2 Comments
Point 1: However, this review is more tend to be literature summarization without attractive opinions and perspectives on the shaping of cancer stem cell plasticity. Beyond that, the author should combined with their own studies on cancer stem cell plasticity to demonstrate review opinions
Response 1: Thank for you suggestive advice. According to your advice, we added the details about our own studies on osteosarcoma stem cells (OSCs). Based on our previous studies, we put forward the opinions about dynamic change between OSCs and non-OSCs (Page 4 line 123-124, Page 9 line 372-374, in yellow).
Point 2: More importantly, if it is available, how about clinical practice on targeting cancer stem cell, and can we apply experiences from other cancer types on the cancer stem cell plasticity?
Response 2: The major aim on CSCs research is to find the specific targets for treating cancer. Several studies reported the drugs targeting CSCs. Fresolimumab (GC-1008) as TGF-β inhibitor is used to treat CSCs and GC-1008 is going on Phaseâ… or â…¡ corresponding for malignant melanoma or metastatic breast cancer. Another drug is Rovalpituzumab tesirine (Rova-T) conjugated DDL3 (an atypical Notch ligand) antibody performing in Phase â…¢ for small-cell lung cancer. Besides, inducing CSCs differentiation is also an efficient strategy, a series of drugs aiming Wnt or Notch signal pathway also have great therapy potential. According to your helpful advice, we added corresponding details in the manuscript (Page 11 line 436-440 and line 443-445, in yellow).

Round 2
Reviewer 2 Report
NA